# Hardware-Assisted Low-Latency NPU Virtualization Method for Multi-Sensor AI Systems

**DOI:** 10.3390/s24248012

**Published:** 2024-12-15

**Authors:** Jong-Hwan Jean, Dong-Sun Kim

**Affiliations:** Department of Semiconductor Systems Engineering, Sejong University, Seoul 05006, Republic of Korea; kingdori1@naver.com

**Keywords:** neural processing unit, virtualization, multi-sensor, hardware scheduler, prefetching

## Abstract

Recently, AI systems such as autonomous driving and smart homes have become integral to daily life. Intelligent multi-sensors, once limited to single data types, now process complex text and image data, demanding faster and more accurate processing. While integrating NPUs and sensors has improved processing speed and accuracy, challenges like low resource utilization and long memory latency remain. This study proposes a method to reduce processing time and improve resource utilization by virtualizing NPUs to simultaneously handle multiple deep-learning models, leveraging a hardware scheduler and data prefetching techniques. Experiments with 30,000 SA resources showed that the hardware scheduler reduced memory cycles by over 10% across all models, with reductions of 30% for NCF and 70% for DLRM. The hardware scheduler effectively minimized memory latency and idle NPU resources in resource-constrained environments with frequent context switching. This approach is particularly valuable for real-time applications like autonomous driving, enabling smooth transitions between tasks such as object detection and route planning. It also enhances multitasking in smart homes by reducing latency when managing diverse data streams. The proposed system is well suited for resource-constrained environments that demand efficient multitasking and low-latency processing.

## 1. Introduction

Single sensors have become more complex and diverse with the rapid development of artificial intelligence in modern society [1]. Now, several functions are integrated into a single sensor to provide various services.

As shown in Figure 1, intelligent sensors are widely used in various areas of our daily lives, and sensor technology continues to advance rapidly [2]. For example, smartwatches have transitioned from simple time-measuring devices to multifunctional tools that include heart rate monitoring and smartphone notification features [3]. Intelligent sensors also play a pivotal role in creating smart homes, where devices like remote controllers, washing machines, robot vacuum cleaners, and air conditioners are interconnected through smartphones. In autonomous driving, sensors such as radar and cameras are used to predict vehicle locations and recognize pedestrians, respectively [4]. Sensors also enable facial recognition for unlocking devices and enhance security through motion detection systems that can respond to movement in real time. In the biomedical field, recent advancements highlight the integration of machine learning (ML) algorithms to improve biosensing accuracy and diagnostic efficiency. AI-based biosensors, for example, support the real-time detection of biomarkers for diseases like cancer and infectious diseases [5,6,7]. Wearable and implantable biosensors leverage ML to reduce latency and enhance decision making in point-of-care (POC) systems [7]. Fast real-time processing has become essential as sensors continue to process increasingly complex data. To meet the growing demands for real-time processing, neural processing units (NPUs) have been combined with sensors to handle complex data in parallel with low power consumption. This integration enables faster execution of multiple tasks, such as real-time noise reduction in image streams for applications like autonomous driving and surveillance [8]. Despite these advancements, reducing latency in environments where multiple sensor applications operate simultaneously remains a critical challenge. NPUs can process deep-learning models at high speed, but when multiple applications concurrently execute different deep-learning models, resource contention often leads to memory bottlenecks, resulting in significant overhead [9]. Traditional software-based NPU virtualization techniques, while addressing some of these challenges, exacerbate latency due to frequent context switching between applications [10]. NeuCloud has attempted to mitigate these issues by focusing on improving resource utilization and ensuring performance isolation in multi-tenant cloud environments [9,10,11]. However, these studies primarily target cloud-based multi-tenant scenarios, leaving gaps in addressing the real-time processing requirements critical for environments such as multi-sensor systems or edge devices.

To address these limitations, this study focuses on reducing real-time processing latency to support multi-sensor AI applications where fast and accurate response times are crucial. We propose a hardware-assisted NPU virtualization method that leverages a hardware scheduler to optimize resource allocation and resolve memory bottlenecks during concurrent task execution. We implemented a cycle accrual NPU virtualization simulator using the DRAMsim3 library to validate our approach. This simulator measures memory cycle values in a virtualized environment where five guest applications run simultaneously. We achieved significant performance improvements by integrating a hardware scheduler into the existing NPU virtualization system and prefetching data from deep-learning layers. Testing various deep-learning models, including CNNs, demonstrated a substantial reduction in memory cycles. These findings highlight the potential of our method to enhance real-time processing in biomedical sensor applications and other real-time critical environments. Moreover, hardware-assisted NPU virtualization maximizes resource utilization and minimizes latency in intelligent multi-sensor systems where real-time processing is essential. Even in resource-constrained, lightweight, and on-device environments, this method significantly improves the performance of deep-learning models [12]. This advancement paves the way for more efficient and responsive AI-driven technologies in fields such as autonomous vehicles, smart cities, advanced robotics, and real-time biomedical applications [12,13].

## 2. Simulation Environment and Methodology

### 2.1. NPU Virtualization Operation Flow

Figure 2 shows the operational flow of NPU virtualization. First, the hypervisor creates guest applications and device drivers. In the generated applications, the applications request vNPU instances from the hypervisor. The hypervisor creates the vNPU instance required for deep-learning of the guest app, such as the number of cores, network configuration, memory size to allocate, input bandwidth, and clock speed. The hypervisor also schedules which guest application should execute first and selects the appropriate vNPU instance to provide [14]. It stores the provided vNPU instance information in control registers to manage resources. Next, the guest application issues commands to start deep-learning model computations. At this stage, the layer data of the deep-learning model are read from DRAM and stored in the on-chip scratchpad memory. The NPU sequentially processes the data by reading the layer data from the scratchpad memory, performing computations, and handling the tasks. Once the tasks are complete, the system saves the final processed layer information, as well as computes cycles, memory addresses, and intermediate results, and the state of the vNPU instance is returned to the hypervisor. The system then initializes the vNPU instance, and the next scheduled guest application begins execution, with the hypervisor allocating the vNPU instance in the same manner as before. This describes the overall operational flow of the NPU virtualization system.

### 2.2. Experimental Setup

We designed this experimental platform to evaluate the performance of NPU virtualization in efficiently executing multiple AI sensor applications. The platform utilizes an enhanced tensor processing unit version 4 (TPUv4) architecture that has been specifically modified to support hardware-assisted NPU virtualization. Through TPUv4, we built an environment to handle deep-learning data from various sensors through NPU virtualization.

#### 2.2.1. Neural Processing Unit (NPU) Architecture

Figure 3 shows the parameters of the TPUv4 architecture set in the simulation. TPUv4 was designed with large-scale computational power to efficiently support demanding AI workloads. TPUv4 features a highly optimized architecture specification, as shown in Table 1. TPUv4 efficiently handles various data using a systolic array (SA) consisting of 128 × 128 processing elements, enabling effective data flow and parallel processing [15]. The SA comprises 16,384 interconnected processing elements (PEs), each directly communicating with its neighboring PEs [16]. This interconnected structure facilitates significantly faster convolution layer computations and real-time data exchange, ensuring optimal performance for deep-learning tasks. For enhanced efficiency, we ensured that only the input and weights were moved using an output stationary method, which fixes the output to enable data reuse and reduce redundant memory operations. We set the tile’s input feature map and the filter size to 786,432 bytes, enabling the division of layers into smaller, manageable chunks for efficient processing [17].

#### 2.2.2. SPM and DRAM Configuration

Table 2 represents the configuration parameter values of the SPM set in the experiment. The parameters were carefully selected to optimize the performance and memory access efficiency of the NPU. First, we set the relevance of the translation lookaside buffer (TLB) to 8 and the total number of entries that can be stored in the TLB to 2048, ensuring efficient virtual-to-physical address translation. The clock speed of both the NPU and DRAM was set to 2 GHz, while the SPM access delay time was configured to one cycle for minimal latency. We configured the ScratchPad memory (SPM) capacity to approximately 37 MB, a value carefully determined based on the memory requirements of all five deep-learning models used in the experiment. Additionally, we set the data block size to 64 bytes, optimizing the memory transactions for each layer’s workload [18].

Table 3 presents the values of the DRAM configuration parameters used in the experiment. We configured the DRAM with eight channels and allocated a channel size of 1024 MB in the simulation. We used a 128-bit bus width, four bank groups, and four banks per group. Each bank was assigned 32,768 rows, and each row was set to have 64 columns. The device width was defined as 128 bits. The Burst Length (BL), which determines the data the DRAM can handle in one operation, was specified as 4 bits. We set the refresh cycle time (tRFC) to 260 cycles and the write recovery time (tWR) to 16 cycles. The clock cycle (tCK) was configured to 1 cycle. The CAS latency (CL), which controls the delay before data are returned after a command, was set to 14 cycles. Similarly, we configured the row-to-column delay (tRCD) to 14 cycles, the row precharge time (tRP) to 14 cycles, and the row active time (tRAS) to 34 cycles. We implemented an open-page policy. Additionally, we set the DRAM operating voltage (VDD) to 1.2 V, the active power consumption (IDD0) to 65 mA, and the current consumption during read operations (IDD4R) to 390 mA [19].

#### 2.2.3. Deep-Learning Models Used

We tested the performance of NPU virtualization using five deep-learning models to simulate different deep-learning models. We tested AlexNet and ResNet-50, which are models designed for the image classification task; NCF, which predicts the right items for users in recommendation systems by learning the interaction between users and items; YOLO-tiny, which is used to quickly and efficiently detect multiple objects in images or videos; and DLRM, which is designed for custom advertisements or product recommendations on large-scale data [20]. We tested these various models and tested the performance of the NPU virtualization system.

### 2.3. NPU Virtualization System

Figure 4 shows the hardware-assisted NPU virtualization system. First, the hypervisor creates a guest app with a device driver, runtime, and deep-learning framework. In addition, it assigns a vNPU instance to the guest application and manages it through the command buffer and control register. The created guest app requests the vNPU resources from the hypervisor to run the deep-learning model, and the hypervisor provides the vNPU resources (as required by the app). In this process, the hypervisor securely accesses the NPU and DRAM memory resources via the IOMMU. When calculating the deep-learning model, the NPU reads the layer’s input and weights from DRAM through the DMA, stores the data in scratch pad memory, and proceeds with the operation. The hypervisor manages the NPU resources through this system so that each guest app can share and use a single NPU resource [21].

#### 2.3.1. Hypervisor Design and Implementation

The hypervisor creates and manages virtual machines and instances and efficiently schedules them so multiple applications can run simultaneously. The hypervisor also interfaces with the memory management unit MMU and the hardware scheduler to efficiently allocate resources. The hypervisor optimizes the scheduling method so context switching between guest apps can occur freely [22]. In context switching, the status of tasks in the suspended deep-learning model is stored in the checkpoint so that functions can resume without data loss at subsequent execution [23]. However, this causes latency and degrades the context switching process of the exchanging data from interrupted tasks [24]. We applied the hardware scheduler to the NPU virtualization system to address this issue. As a result, the hardware scheduler significantly reduced the latency during context switching by quickly recovering data. In addition, we used hardware schedulers as they save scheduling information and algorithms in advance to promptly provide the resources needed for each application in real time, thereby minimizing latency.

#### 2.3.2. Data Prefetching Algorithms via Hardware Scheduler

The hypervisor allows the scheduler to calculate the number of SA required for each layer for each application and allocate these resources as efficiently as possible based on the number of guest applications entered and the maximum number of SA available. It also ensures smooth context switching between guest apps. We used a layer-based approach instead of traditional round-robin or priority-based scheduling methods when conducting the experiments [25]. The scheduler calculates the number of active processing elements (PEs) and SA required for each layer. It groups the layers into one group with as many available resources as possible to allocate SA. The scheduler saves the current state of the application so that it can resume the next interrupted task when the requested SA exceeds the number of available resources. However, the small number of limited SA resources results in significant overhead due to frequent context switching. Algorithm 1 illustrates a prefetching algorithmic system that we introduced to optimize data flow between memory and NPU, allowing deep-learning models to be read from DRAM through hardware during the computation time, thereby minimizing resource idle and improving the processing speed. This algorithm stores scheduling information in the hardware scheduler to precisely coordinate memory operations and computational tasks, reducing unnecessary delays. Hardware-assisted virtualization systems enable faster data exchange and address persistent software-based scheduling challenges. Once an app starts computation, it prefetches data from the layer to the possible layer running on the next app. It is based on scheduling information within the hardware scheduler’s acceptable memory and within the calculation time of the app running before context switching. Thus, hardware schedulers speed up each guest application’s transition and minimize unused time resources. They also enable more efficient multitasking in multi-tenant environments. The proposed NPU virtualization system optimizes task execution and resource utilization through an efficient data prefetching algorithm [26]. The system uses a layer-based approach to ensure efficient task processing in resource-constrained environments.
**Algorithm 1:** Data Prefetching for Hardware-Assisted NPU Virtualization
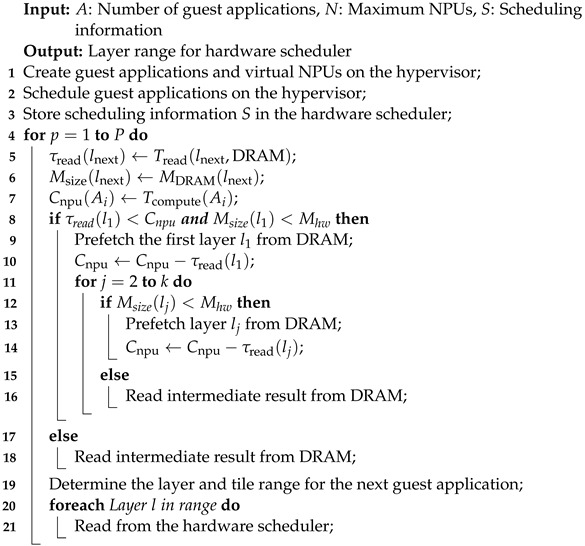


During initialization, the hypervisor identifies the number of guest applications and the maximum available SAs (systolic arrays). The hardware scheduler receives SA count information for each application required per layer from the hypervisor, determines the range of layers that can be processed, and checks for prefetch availability. The system calculates the number of SAs needed for each layer, ensuring resource allocation is optimized even in resource-constrained environments. This allows the scheduler to handle as many deep-learning model layers as possible by leveraging the maximum resources the hypervisor can provide to guest apps. The scheduler stores the application’s state (current layer, computational cycle, etc.) such that the deep-learning model can resume usually later. The scheduler continues the aborted task when that deep-learning model is re-run later. This process continues up to the available memory capacity $M_{\mathrm{hw}}$ of the hardware scheduler. The system loads data from subsequent operations into the available memory by prefetching data from the following application during the current computation, thus reducing idle time during context switching. By allowing memory access and computation to proceed simultaneously, resource waste is minimized and overhead is reduced. The data prefetching algorithm dynamically reads the layer data from the DRAM during computation and stores it in the hardware scheduler memory, making real-time processing more efficient and adaptable to varying workloads. The hardware scheduler computes three important parameters: the memory size of the layer $M_{\mathrm{size}}$, the time to read that memory of the layer $T_{\mathrm{read}}$, and the time to compute a deep-learning model $C_{\mathrm{npu}}$ before a context switching. The system prevents the NPU from idling while waiting for memory access by pre-processing the process of importing layers from the deep-learning model when the next application runs. Specifically, the algorithm calculates the memory size to be imported from the DRAM by calculating the read time $T_{\mathrm{read}}$ and the memory size $M_{\mathrm{size}}$. In the hardware scheduler, if $M_{\mathrm{next}}$ has less memory for a layer to be imported from the DRAM than the available memory $M_{\mathrm{hw}}$, then that layer is prefetched to the hardware scheduler to reduce memory access time. Here, $M_{\mathrm{DRAM}}\left(l_{\mathrm{next}}\right)$ is a function used to calculate the memory size $M_{\mathrm{size}}$ of layer $l_{\mathrm{next}}$. If the hardware scheduler has enough memory to store the layer, the algorithm continues prefetching the layer data while the layer operation of the previous application is in progress. The scheduler dynamically manages the balance between prefetching and computation to ensure that the NPU is not idle. In addition, to ensure accurate layer prefetching time, $T_{\mathrm{read}}$ is continuously updated to check if prefetching is still possible. When time or memory constraints make prefetching impossible, the system interrupts the process and stores several operational states, such as the computational cycle of the current layer. This process allows us to seamlessly resume work from the saved state when the next context switching occurs, ensuring multiple applications can effectively manage resources. For example, $C_{\mathrm{npu}}\left(A_{i}\right)$ represents the NPU computational cycle required by an application, and $T_{\mathrm{compute}}\left(A_{i}\right)$ is a function that calculates the total NPU computational cycles for that application. These variables determine the number of layers to be stored in the hardware scheduler from the deep-learning model of the following application, allowing flexible handling of multiple applications with less overhead. This layered optimization minimizes unnecessary delays, enhancing multitasking efficiency in dynamic environments. Even in cases of frequent context switching, the idle time of NPU resources is minimized.

## 3. Results

First, scheduling was repeated from Alexnet to Resnet-50, NCF, Yolo-tiny, and DLRM to run with context switching until all models were complete. We set the hardware scheduler’s memory to 1 MB.

### 3.1. Memory Access Cycles Under Different Burst Sizes and SA Counts

This section compares and analyzes the total memory cycles before and after applying the hardware scheduler. The experiments were conducted by changing the values of parameters such as burst size (64 bytes and 128 bytes) and the maximum available SA. The evaluation model used five deep-learning models: AlexNet, ResNet-50, NCF, Yolo-tiny, and DLRM.

#### 3.1.1. Effect of the Hardware Scheduler by Changing Burst Size

Figure 5 presents the experimental outcomes obtained by configuring the burst size to 64 and 128 bytes. When applying the hardware scheduler, memory cycles were significantly reduced when the burst size was 64 and 128 bytes.

The total memory cycle of AlexNet decreased by about 12.63% when the burst size was 64 bytes, as shown in Table 4. ResNet-50, NCF, and Yolo-tiny also showed reductions of 14.24%, 36.38%, and 17.07%, respectively. DLRM showed a 76.82% decrease due to the hardware scheduler as most memory accesses were performed within the computational time of previous applications, and most of the memory accesses were performed within the computational time of prior applications.

According to Table 5, when the burst size was 128 bytes, applying the hardware scheduler resulted in a 13.96% reduction in the memory cycles of AlexNet. At the same time, ResNet-50 saw a reduction of 15.48%, NCF saw 36.46%, Yolo-tiny saw 15.95%, and DLRM saw 83.28%. In both cases, hardware schedulers were used to significantly reduce the memory cycle, and the effectiveness of the hardware scheduler was significant even when the burst size was changed. Therefore, applying NPU virtualization systems with hardware schedulers to intelligent multi-sensor systems enables faster and more efficient complex data processing.

#### 3.1.2. Effect of Hardware Scheduler by Changing a Limited Number of SA Resources

Next, experiments were conducted by limiting the number of available SA resources. Figure 6 and Table 6, Table 7 and Table 8 show the changes in memory cycles for reading layers when the number of SA resources was 30,000, 40,000, and 50,000, respectively, for each model before and after applying the hardware scheduler. Figure 7 illustrates the differences in memory cycles by subtracting the memory cycles after applying the hardware scheduler from those before its application. First, for 30,000, Alexnet, Resnet-50, NCF, Yolo-tiny, and DLRM showed reductions of 12.63%, 14.24%, 36.38%, 17.07%, and 76.82%, respectively, before and after the hardware scheduler was applied. For 40,000, Alexnet, Resnet-50, NCF, Yolo-tiny, and DLRM showed reductions of 8.33%, 11.92%, 33.19%, 11.31%, and 73.22%, respectively. Next, for 50,000, Alexnet, Resnet-50, NCF, Yolo-tiny, and DLRM showed reductions of 3.16%, 8.14%, 30.86%, 9.77%, and 71.7%, respectively. As shown in Figure 7, the effectiveness of the hardware scheduler was noticeable with a smaller limited number of SA resources, with each deep-learning model showing a reduction of almost over 12% or more when the limited number of SA was 30,000. For DLRM models with less memory access time than other deep-learning models, the memory access time was reduced by more than 70% in all cases. The DLRM model has relatively low memory requirements, and the hardware scheduler was able to prefetch a significant amount of data needed during the computational time of the previous model in the hardware scheduler’s on-chip memory.

The impact of the scheduler was more evident with a smaller number of SA available at 30,000, but performance improvements were still observed with an increase to 50,000. This allows for more efficient context switching in the frequent context switching with NPU virtualization. In conclusion, we show that NPU virtualization systems with a hardware scheduler can use limited resources more efficiently to minimize memory access time and handle layers from multiple deep-learning models faster. As a result, NCF and DLRM models have seen the most significant reduction in latency through the scheduler, and the scheduler’s ability to prefetch data and reduce memory overhead is highly effective during frequent context switching.

## 4. Conclusions

In this study, we optimized NPU resource allocation in intelligent multi-sensor systems and significantly reduced memory access overhead through hardware-assisted NPU virtualization. Additionally, we reduced the memory stall time through using a prefetching algorithm via the hardware scheduler, allowing each application to efficiently process deep-learning model layers. The prefetching algorithm minimized idle NPU resource states by preloading the deep-learning model layers running continuously in the DRAM into the hardware scheduler. It also significantly improved the multitasking processing speed and performance. When the memory space was limited, the DLRM and NCF models significantly reduced memory access overhead when compared to other deep-learning models. This was because the computation time of the previous deep-learning model before context switching was sufficient, and the memory size of each layer in DLRM and NCF was smaller compared with other models. The DLRM model, in particular, had relatively small layer sizes compared with the different deep-learning models in the experiment. This allowed most of the necessary layer data to be preloaded from DRAM into the hardware scheduler during the previous task’s computation time. As a result, with 30,000 SA resources, memory access cycles were reduced by up to 76%. The NCF model also showed a reduction of over 30% in memory cycles in each case. Moreover, when we limited the SA resources to 30,000, AlexNet decreased by 12%, ResNet-50 by 14%, and YOLO-tiny by 17%. These results demonstrate that hardware-based virtualization processes deep-learning models significantly faster than traditional software-based virtualization systems, achieving performance improvements of over 10% across various deep-learning models. The DNN models tested in this study—AlexNet, ResNet-50, YOLO-tiny, DLRM, and NCF—were selected for their suitability in specific application domains. AlexNet is well suited for the lightweight image classification tasks often required in resource-constrained edge environments and wearable devices thanks to its relatively small model size that enables efficient operation in such scenarios. ResNet-50 excels at extracting fine-grained details from high-resolution images, making it ideal for applications such as medical imaging and precision surveillance. YOLO-tiny provides real-time object detection capabilities, making it particularly valuable in dynamic environments like autonomous driving and security systems that require low-latency processing. DLRM is optimized for handling large-scale datasets in applications such as social networks and recommendation systems, offering low memory requirements and high processing efficiency. NCF is well suited for modeling relationships between users and items, making it highly effective in personalized recommendation systems. These models were used to validate the versatility and efficiency of the hardware scheduler in supporting various applications, demonstrating exceptional performance in their respective application scenarios. This approach demonstrates the potential for broader adoption in resource-constrained environments requiring real-time and efficient multi-task processing. While existing studies have mainly proposed new scheduling methods based on software, this study introduced hardware schedulers to validate significant performance improvements. The hardware scheduler minimized memory access overhead in multi-tenant environments, optimizing resource allocation when loading layer data into the NPU. The effectiveness of the hardware scheduler became more prominent with fewer resources and more frequent context switching. Modern lightweight or on-device AI systems often need to simultaneously execute diverse applications with limited resources, although the benefits of the hardware scheduler may be limited in environments with minimal app transitions [27,28,29]. In such cases, the hardware scheduler is expected to achieve significant results. Furthermore, this approach is particularly beneficial in specific applications such as medical research, security, and surveillance. For instance, in medical research, wearable or implantable medical devices can handle complex and diverse data streams with fast transitions and processing while reducing latency in real-time biosensing and diagnostics. In security and surveillance systems, the proposed method accelerates the processing of multiple camera feeds and sensor data, contributing to improved real-time detection capabilities. This approach also reduces power consumption and ensures more efficient use of resources by utilizing a single NPU to simultaneously run multiple applications. Additionally, this method facilitates faster and more cost-effective data processing in real-time multi-tenant environments, enhancing the practicality and applicability of this approach.

## Figures and Tables

**Figure 1 sensors-24-08012-f001:**
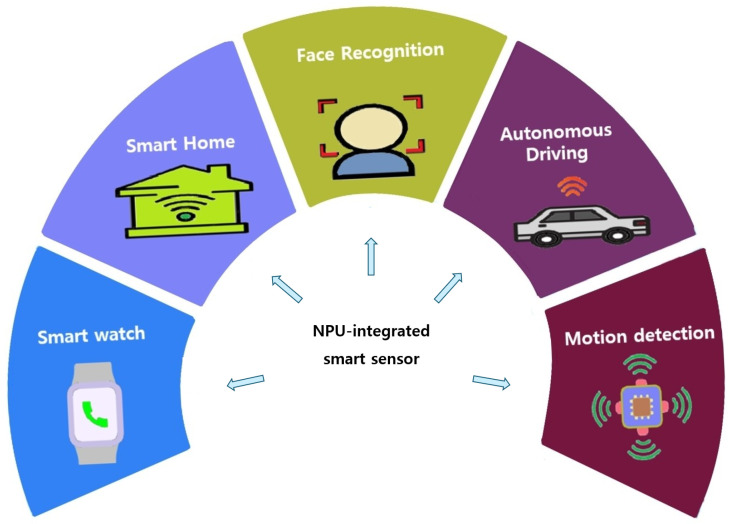
Examples of real-life applications of multi-sensor AI in various fields.

**Figure 2 sensors-24-08012-f002:**
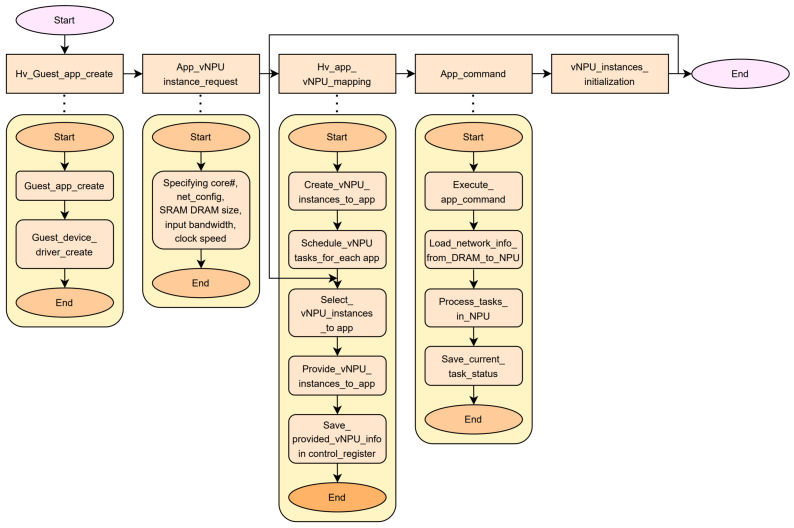
NPU virtualization operation flow. The symbol ‘#’ represents the number of cores.

**Figure 3 sensors-24-08012-f003:**
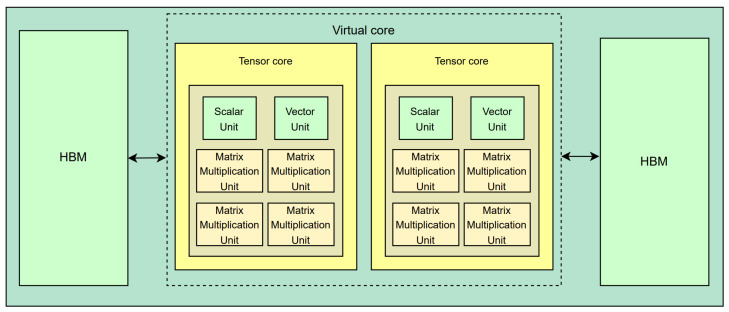
TPU v4 architecture.

**Figure 4 sensors-24-08012-f004:**
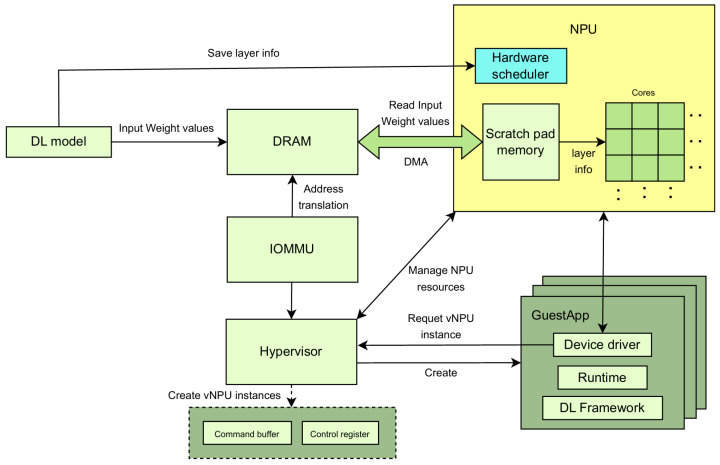
Hardware-assisted NPU virtualization system.

**Figure 5 sensors-24-08012-f005:**
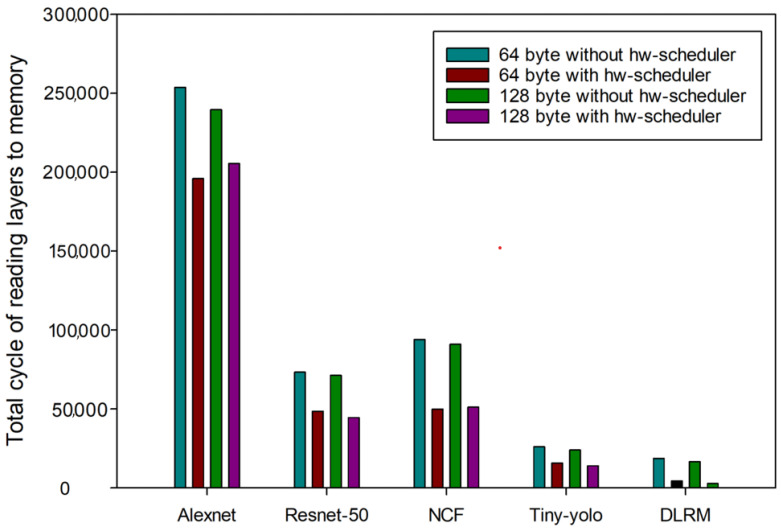
Comparison before and after the hardware scheduler when the burst size was changed.

**Figure 6 sensors-24-08012-f006:**
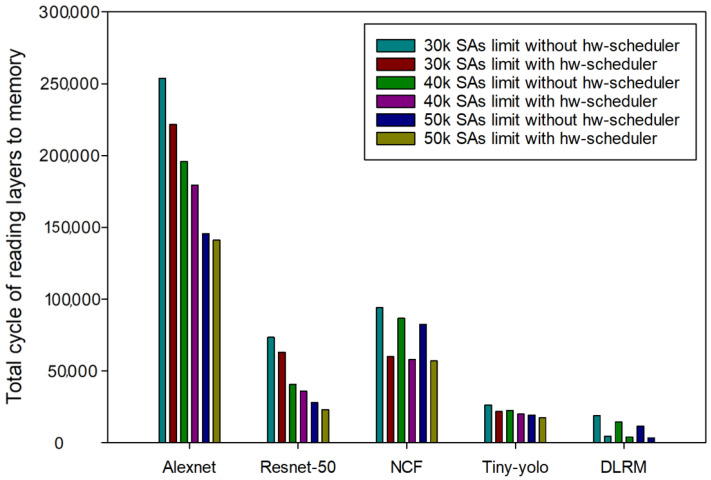
Comparison before and after hardware scheduler application when the number of available SA changed.

**Figure 7 sensors-24-08012-f007:**
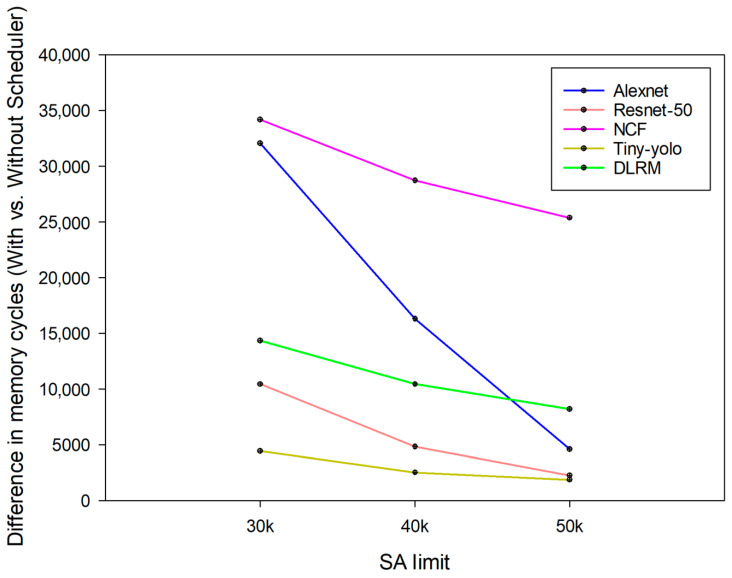
Difference in memory cycles with and without a hardware scheduler.

**Table 1 sensors-24-08012-t001:** TPUv4 configuration parameters.

Parameter	Value
Data flow type	Output stationary
Systolic height	128
Systolic width	128
Tile ifmap size (byte)	786,432
Tile filter size (byte)	786,432
Tile ofmap size (byte)	786,432

**Table 2 sensors-24-08012-t002:** SPM configuration parameters.

Parameter	Value
Tlb assoc	8
Tlb entrynum	2048
Npu clock speed (GHz)	2
Dram clock speed (GHz)	2
SPM size (bytes)	37,748,736
SPM latency	1
Data block size (bytes)	64

**Table 3 sensors-24-08012-t003:** DRAM configuration parameters.

Parameter	Value
Channels	8
Bus Width (bit)	128
Bank Groups	4
Banks per Group	4
Rows per Bank	32,768
Columns per Row	64
Device Width (bit)	128
Burst Length (BL)	4
tCK (ns)	1
CL (CAS Latency)	14
tRCD (Row-to-Column Delay)	14
tRP (Row Precharge Time)	14
tRAS (Row Active Time)	34
tRFC (Refresh Cycle Time)	260
tWR (Write Recovery Time)	16
VDD (V)	1.2
IDD0 (Active Power)	65 mA
IDD4R (Read Power)	390 mA
Channel Size (MB)	1024
Row Buffer Policy	Open Page

**Table 4 sensors-24-08012-t004:** Memory cycle reduction (burst size: 64 bytes) with a hardware scheduler.

Model	1-Before	2-After	Reduction (%)
Alexnet	253,764.0000	221,712.0000	12.63%
Resnet-50	73,282.0000	62,847.0000	14.24%
NCF	93,940.0000	59,763.0000	36.38%
Yolo-tiny	25,996.0000	21,558.0000	17.07%
DLRM	18,656.0000	4324.0000	76.82%

**Table 5 sensors-24-08012-t005:** Memory cycle reduction (burst size: 128 bytes) with a hardware scheduler.

Model	1-Before	2-After	Reduction (%)
Alexnet	239,652.0000	206,201.0000	13.96%
Resnet-50	71,220.0000	60,198.0000	15.48%
NCF	91,020.0000	57,837.0000	36.46%
Yolo-tiny	24,046.0000	20,211.0000	15.95%
DLRM	16,521.0000	2762.0000	83.28%

**Table 6 sensors-24-08012-t006:** Memory cycle reduction with hardware scheduler for SA limits (30,000 SA).

Model	30,000 SA Before	30,000 SA After	Reduction (%)
Alexnet	253,764.0000	221,712.0000	12.63%
Resnet-50	73,282.0000	62,847.0000	14.24%
NCF	93,940.0000	59,763.0000	36.38%
Yolo-tiny	25,996.0000	21,558.0000	17.07%
DLRM	18,656.0000	4324.0000	76.82%

**Table 7 sensors-24-08012-t007:** Memory cycle reduction with hardware scheduler for SA limits (40,000 SA).

Model	40,000 SA Before	40,000 SA After	Reduction (%)
Alexnet	195,700.0000	179,400.0000	8.33%
Resnet-50	40,473.0000	35,650.0000	11.92%
NCF	86,524.0000	57,811.0000	33.19%
Yolo-tiny	22,326.0000	19,802.0000	11.31%
DLRM	14,254.0000	3816.0000	73.22%

**Table 8 sensors-24-08012-t008:** Memory cycle reduction with hardware scheduler for SA limits (50,000 SA).

Model	50,000 SA Before	50,000 SA After	Reduction (%)
Alexnet	145,632.0000	141,027.0000	3.16%
Resnet-50	27,889.0000	25,620.0000	8.14%
NCF	82,242.0000	56,864.0000	30.86%
Yolo-tiny	19,124.0000	17,256.0000	9.77%
DLRM	11,412.0000	3224.0000	71.7%

## Data Availability

The deep-learning model data used in this study can be found at the following URL https://github.com/casys-kaist/mNPUsim (accessed on 12 December 2024).

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
