# Peer review of "Hardware-Assisted Low-Latency NPU Virtualization Method for Multi-Sensor AI Systems"

_sensors, 2024, doi:10.3390/s24248012_

Round 1

Reviewer 1 Report

Comments and Suggestions for Authors

It is interesting in the manuscript to reduce processing time and improving resource utilization through hardware schedulers by allowing multiple deep-learning models to be processed simultaneously through the virtualization of NPU. It will be better to show the difference in figure form when the hardware schedulers are used and not. And the training process can be shown and analysed to verify the effect of the proposed method.

Comments on the Quality of English Language

The English express could be improved to make the manusript better.

Reviewer 2 Report

Comments and Suggestions for Authors

In my opinion the manuscript is well written and minor additions are required for publication. here are my suggestions

1. The abstract must clearly narrate (quantitatively if possible) all the major benefits of using this scheme such as processing time, accuracy, efficiency and ease of implementation.

2. Some of the possible state of the art applications of this work must be highlighted in the abstract.

3. In the literature review some emerging trends of using ML algorithms in biomedical research should be included

https://www.sciencedirect.com/science/article/pii/S2666831923000875 https://www.mdpi.com/2079-6374/12/12/1181 https://www.mdpi.com/2075-4418/14/11/1100   4.Results must clearly narrate the Pros and Cons of this work and  in specific applications such as medical research, security and surveillance.  5. Sensor type and specifications must be included if any of them is tested experimentally 6. In conclusions the application based appropriateness of AlexNet, ResNet and YOLO-tiny methods must be explicitly examined.  

Reviewer 3 Report

Comments and Suggestions for Authors

  1. The authors did not provide a review of the state of the art in the area of NPU virtualization research and following this, do not provide any comparisons in their results with existing works in the field.

  2. Figure 4 and 5 contain spelling errors.

  3. Figure 4 does not describe any new mechanisms not shown in Figure 5. Figure 5 should be used in describing the mechanisms in Section 2.3.

  4. The authors did not provide a conclusion section to the paper. The discussion section should be separated from the conclusion, which should summarize the outcomes of the research and the potential for future work.

Round 2

Reviewer 3 Report

Comments and Suggestions for Authors

The Authors have amply revised the manuscript and it can be accepted in its current format